# To maximize or randomize? An experimental study of probability matching in financial decision making

Andrew W. Lo[1,2,3,4☯], Katherine P. Marlowe[2☯], Ruixun Zhang[2☯]*

**1** MIT Sloan School of Management, Cambridge, MA, United States of America, **2** MIT Laboratory for Financial Engineering, Cambridge, MA, United States of America, **3** MIT Computer Science and Artificial Intelligence Laboratory, Cambridge, MA, United States of America, **4** Santa Fe Institute, Santa Fe, NM, United States of America

☯ These authors contributed equally to this work.
* rxzhang@alum.mit.edu

**Data Availability Statement:** The full data is publicly available at: https://figshare.com/s/4760fc73b07ce32b5fe5.

**Funding:** Research support from the MIT Laboratory for Financial Engineering is gratefully

## Abstract

Probability matching, also known as the "matching law" or Herrnstein's Law, has long puzzled economists and psychologists because of its apparent inconsistency with basic self-interest. We conduct an experiment with real monetary payoffs in which each participant plays a computer game to guess the outcome of a binary lottery. In addition to finding strong evidence for probability matching, we document different tendencies towards randomization in different payoff environments—as predicted by models of the evolutionary origin of probability matching—after controlling for a wide range of demographic and socioeconomic variables. We also find several individual differences in the tendency to maximize or randomize, correlated with wealth and other socioeconomic factors. In particular, subjects who have taken probability and statistics classes and those who self-reported finding a pattern in the game are found to have randomized more, contrary to the common wisdom that those with better understanding of probabilistic reasoning are more likely to be rational economic maximizers. Our results provide experimental evidence that individuals—even those with experience in probability and investing—engage in randomized behavior and probability matching, underscoring the role of the environment as a driver of behavioral anomalies.

## 1 Introduction

Most economic theories are based on the premise that individuals maximize their self-interest and correctly incorporate the structure of their environment into their decisions. This framework has led to numerous advances, including expected utility theory [1], game theory [1, 2], rational expectations [3], the efficient markets hypothesis [4, 5], and option pricing theory [6, 7]. The influence of this paradigm goes far beyond academia; it underlies current macroeconomic and monetary policy making, becoming an integral component of the rules and regulations that govern financial markets today [8, 9].

acknowledged. The funders had no role in study design, data collection and analysis, decision to publish, or preparation of the manuscript.

**Competing interests:** The authors have declared that no competing interests exist.

However, there is mounting empirical and experimental evidence suggesting that humans do not always behave in the way traditional economic models predict, but often make seemingly random and suboptimal decisions [10]. These behavioral anomalies and psychological traits are especially pronounced when elements of risk and probability are involved. Examples include loss aversion [11–14], overconfidence [15, 16], overreaction [17], herding [18] psychological accounting [19], miscalibration of probabilities [20], the uncertainty effect [21], and confirmation bias [22]. The spectacular rise of US stock market prices during the tech bubble in the early 2000s, and the even more spectacular crash following the 2007–2008 financial crisis, has intensified the controversy surrounding the rationality of investors.

One particularly interesting behavioral anomaly is probability matching, also known as the "matching law," or Herrnstein's Law [23–30]—the tendency of the relative frequency of predictions of outcomes of an independent randomized event to match its underlying probability distribution. The best-known example of probability matching is the human tendency to choose randomly between heads and tails when asked to guess the outcomes of a series of biased coin tosses. When individuals are asked to guess the repeated outcomes of a biased coin, say with a bias of 70% heads, and rewarded based on whether they guessed correctly, most subjects seem to randomize their guesses at around 70% heads, instead of engaging in the economically optimal behavior of always guessing heads.

Probability matching has long puzzled economists and psychologists because of its apparent inconsistency with basic self-interest. The idea of randomizing behavior is especially difficult to reconcile with the standard economic paradigm of expected utility theory, in which individual behavior is non-stochastic and completely determined by the individual's utility function, budget constraints, and the probability laws governing the environment. For example, Kogler and Kuhberger [31] report that, "Experimental research in simple repeated risky choices shows a striking violation of rational choice theory: the tendency to match probabilities by allocating the frequency of response in proportion to their relative probabilities".

Nevertheless, probability matching has been observed in thousands of geographically diverse human subjects over several decades, as well as in other animal species, including ants [32–35], bees [36–38], fish [39, 40], pigeons [41, 42], and primates [43]. In virtually any setting where an animal is able to make a choice between A versus B in a randomized experiment, we observe probability matching.

The source of these irrational behaviors is often attributed to psychological factors, such as fear, greed, and other emotional responses. However, the fact that some of these behaviors are observed so consistently across species suggests that they may have a more fundamental and common origin, one with an evolutionary role that belies their apparent shortcomings. For example, the neurological basis of probability matching has been investigated extensively [44–49]. In the context of a binary choice model, Brennan and Lo [50] show that probability matching behavior is perfectly consistent with evolution, arising purely from the forces of natural selection and population growth. Moreover, under generalized environmental conditions, i.e., broad assumptions about the conditions required for reproductive success, they derive more general types of behavior that involve randomization, but not necessarily strict probability matching.

In this paper, we present the first experimental test of the evolutionary model of Brennan and Lo [50]. We design an experiment in real-world decision making with monetary payoffs to measure the degree of probability matching among individuals, its determining factors, and its level of variation. Here by probability matching we mean the "matching law," or Herrnstein's Law discussed above—the tendency to choose randomly between heads and tails when asked to guess the outcomes of a series of biased-coin tosses, where the randomization frequency matches the probability of the biased coin.

We recruited a sample of 82 volunteers from the MIT Behavioral Research Laboratory to participate in our experiment. Each participant played a computer game consisting of 200 trials of a binary choice decision. In each trial, an image of either Angelina Jolie or Brad Pitt was displayed with a certain probability, and subjects were paid according to the number of trials in which they correctly guessed which image appeared.

By varying the payoff structure of the game, we were able to test whether subjects showed probability matching behavior, and whether deviations occurred as predicted by the model in Brennan and Lo [50]. Specifically, we designed several payoffs where the evolutionarily dominant behavior was either to maximize, i.e., always to choose one option, or to randomize, i.e., to choose randomly between two options. We found strong evidence for a behavioral difference between theoretical maximizers and theoretical randomizers, as predicted by Brennan and Lo [50]. After controlling for a wide range of demographic and socioeconomic variables, theoretical randomizers still engaged in randomizing behavior more often than theoretical maximizers. When facing different environments (i.e., payoffs in the experiment), our subjects responded differently by adapting to the new conditions and showing different stable behaviors.

We were also able to study individual differences in the tendency to maximize or randomize by collecting basic demographic and socioeconomic information from the anonymous participants. We found that subjects with a higher level of financial assets tended to randomize less often, while subjects with children tend to randomize more often. Moreover, subjects who had taken probability and statistics classes and those who self-reported finding a pattern in the game (none existed) also tended to randomize more often, contrary to our prior expectation that those participants with a better understanding of probability might be more likely to adopt the economically maximizing behavior. In fact, we found that those subjects engage in the exact opposite behavior. This may be due to an attempt to "beat the game," based on the qualitative answers to our post-trial survey by participants.

From the evolutionary perspective, the key to understanding these behavioral predictions lies in the assumption of systematic reproductive risk [50, 51]. The experiment we describe in this article involves a binary choice in which the risks to the population are idiosyncratic, that is, the outcomes of one individual's choice are independent of those of another. However, when individuals with preferences formed in response to systematic risks are placed in the different environment, there is the potential for probability matching to occur, creating what appears to be irrational behaviors for those environments.

Our results contribute to the growing literature on rationalizing the existence of probability matching. As far back as the 1950s, researchers [52–54] developed statistical models that attempted to explain and predict matching behavior. Since then, several behavioral reasons have been offered, including its emergence as a consequence of pattern searching [55], through the greater utility gained from guessing the rarer event correctly [56], and by the role of diversification to avoid boredom [57]. More recently, explanations of probability matching have been proposed from an evolutionary point of view. Wolford, Miller, and Gazzaniga [45] argue that early human beings look for explanatory causal relationships as a survival strategy. Wozny, Beierholm, and Shams [58] have shown that humans match probabilities not only in cognitive tasks, but also in perceptual tasks. This implies that the human nervous system has a built-in function that samples from a distribution of hypotheses, and updates its belief after each observation.

Our results provide experimental validation for the predictions of Brennan and Lo [50], as well as additional evidence that individuals engage in randomized behavior and probability matching, even those with prior experience in probability and investing. More importantly, our results may provide an explanation for several notable departures from exact probability

matching [31, 59, 60]. Randomizing behavior that matches environmental probabilities depends on the relative reproductive success of the outcomes, and the evolutionary framework proposed in Brennan and Lo [50] offers a simple and specific set of conditions for understanding and predicting such behavior.

## 2 Materials and methods

### 2.1 Evolutionary origins of probability matching

Brennan and Lo [50] proposed an evolutionary framework for the origin of several behaviors that are considered "anomalous" in economic theories based on the assumption of rational behavior. In particular, probability matching—the tendency of the relative frequency of guesses of the outcomes of a sequence of independent random events to match the underlying probability distribution of events—can be explained when the uncertainty in environment is systematic across all individuals, an example demonstrating that natural selection is able to yield behaviors that may be individually sub-optimal but are optimal for the population. For expositional convenience, we present a brief review of this framework here, and then turn to our experimental design.

We begin with a population of individuals that live for one period, produce a random number of offspring asexually, and then die. During their lives, individuals make only one decision: they choose one of two possible courses of action, denoted $a$ and $b$, and this choice results in one of two corresponding random numbers of offspring, $x_a$ and $x_b$, given by:

$$
\begin{aligned}
\mathrm{Prob}(x_a = c_{a1}, x_b = c_{b1}) &= p \in [0,1] \\
\mathrm{Prob}(x_a = c_{a2}, x_b = c_{b2}) &= 1 - p \equiv q
\end{aligned}
\tag{1}
$$

where $p$ is some probability between 0 and 1.

We further assume that $x_a$ and $x_b$ are independently and identically distributed over time, and identical for all individuals in a given generation. In other words, if two individuals choose the same action $a$, both will produce the same number of random offspring $x_a$. This implies that the variation in offspring due to behavior is wholly systematic, i.e., the link between action and reproductive success is the same throughout the population.

A "mindless" individual's behavior in this world is fully specified by the probability of choosing action $a$. Following the notation in Brennan and Lo [50], we denote this probability as $f$. Each individual dies after one period, and we assume its behavior $f$ is heritable: offspring will behave in a manner identical to their parents, i.e., they choose between the two actions according to the same probability $f$.

From the individual's perspective, always choosing the action with a higher expected reproductive success ($f$ = 0 or 1) will lead to more offspring on average. However, Brennan and Lo [50] showed that from the perspective of the population, this individually optimal behavior cannot survive. In fact, the evolutionarily dominant behavior will depend on the relationship between the probability $p$ and the relative fecundity ratios $r_j := c_{aj}/c_{bj}$ for each of the two possible states of the world, $j = 1, 2$, where $f$ can be anywhere between 0 and 1 in general, implying randomized behavior. See Proposition 3 of Brennan and Lo [50] for more detail.

Fig 1 illustrates the evolutionarily dominant behavior $f^*$ as a function of $r_1$ and $r_2$. If $r_1$ and $r_2$ are not too different in value—i.e., the ratio of fecundity between choices $a$ and $b$ is not very different between the two states of the world—then random behavior yields no evolutionary advantage over deterministic choice. In this case, the individually optimal behavior ($f^*$ = 0 or 1) will prevail in the population.

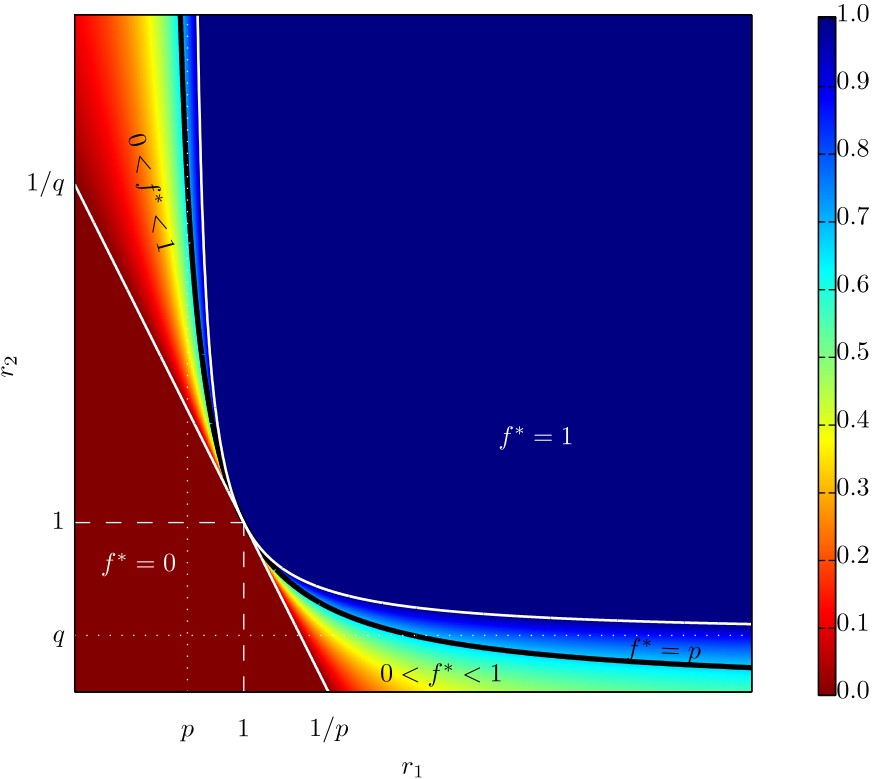

**Fig 1. Regions of the $(r_1, r_2)$-plane that imply deterministic ($f^* = 0$ or 1) or randomizing ($0 < f^* < 1$) behavior**, where $r_j = c_{aj}/c_{bj}$ measures the relative fecundities of action $a$ to action $b$ in the two states $j = 1, 2$. The asymptotes of the curved boundary line occur at $r_1 = p$ and $r_2 = q$. Values of $r_1$ and $r_2$ for which exact probability matching is optimal are given by the solid black curve. Source: Brennan and Lo [50, Fig 1].

However, if one of the $r$ variables is large while the other is small, then random behavior will be more advantageous for the population than a deterministic one. In such cases, there are times in which each choice performs substantially better than the other. Under those conditions, it is evolutionarily optimal for a population to diversify between the two choices, rather than always choosing the outcome with the highest probability of progeny in a single generation.

A simple numerical example from Brennan and Lo [50] will illustrate the basic mechanism of this model. Consider a population of individuals, each facing a binary choice between one of two possible actions, $a$ and $b$. 70% of the time, environmental conditions are positive, and action $a$ leads to reproductive success, generating 3 offspring for the individual. 30% of the time, environmental conditions are negative, and action $a$ leads to 0 offspring. This corresponds to $p = 70\%$, $c_{a1} = 3$, $c_{b1} = 0$ in the notation of (1). Suppose action $b$ has exactly the opposite outcomes—whenever $a$ yields 3 offspring, $b$ yields 0, and whenever $a$ yields 0, $b$ yields 3. This corresponds to $c_{a2} = 0$, $c_{b2} = 3$ in the notation of (1). From the individual's perspective, always choosing $a$, which has the higher probability of reproductive success, will lead to more offspring on average. However, if all individuals in the population behaved in this "rational" manner, the first time that a negative environmental condition occurs, the entire population will become extinct. Assuming that offspring behave identically to their parents, the behavior "always choose $a$" cannot survive over time. For the same reason, "always choose $b$" is also unsustainable. In fact, one can show that in this special case, the behavior with the highest reproductive success over time is for each individual to choose $a$ 70% of the time and $b$ 30% of

the time, matching the probabilities of reproductive success and failure. Eventually, this particular randomizing behavior will dominate the entire population.

The key to understanding these behavioral predictions lies in the assumption of *systematic* reproductive risk. This dependence on risk has implications that go far beyond the current setting. For example, Zhang, Brennan, and Lo [51] show that environments with a mix of systematic and idiosyncratic reproductive risks cause different risk preferences to emerge. While our risk preferences may be determined by the nature of the risks to which we and our evolutionary ancestors have been exposed, we do not necessarily have the ability to distinguish between systematic and idiosyncratic risks in our day-to-day decision making.

## 2.2 The binary choice game

Turning to our experimental design, we presented live human subjects with a binary choice game in which the risks to the population are idiosyncratic, that is, the outcomes of one individual's game are independent of those of another. However, when individuals apply preferences formed in response to systematic risks to the wrong environment, there is the potential for probability matching to occur, creating what appears to be irrational behaviors for those environments.

In our experiment consisted of four particular payoff structures by varying the parameters in (1) (equivalently, four particular points in Fig 1), and observe whether participants show behaviors predicted by Brennan and Lo [50] (equivalently, behaviors indicated by different colors in Fig 1).

We recruited a sample of 82 volunteers and conducted our binary choice experiment at the MIT Behavioral Research Laboratory. Our subjects were varied in their personal and socioeconomic characteristics. We provide a summary of their statistics in Section 3.1.

The full experimental session typically lasted 45 to 60 minutes for a given participant. Each participant used a computer program that completed 200 iterations of a binary choice trial, in essence playing a lottery. On each iteration of the trial, subjects were shown an image of one of two popular film stars—Angelina Jolie or Brad Pitt—with specific fixed probabilities that were unknown to the subjects. Each participant was randomly assigned to one of four experimental designs, as shown in Table 1. In designs 1 and 2, Angelina Jolie appeared 70% of the time and Brad Pitt 30% of the time. Designs 3 and 4 used the opposite probabilities. The participant guessed which image would appear before it was revealed, and the participant would receive a certain amount of virtual dollars if their guess was correct. Fig 2 shows a screenshot of the computer interface used in the experiment.

In designs 1 and 4, subjects received two virtual dollars when they guessed correctly, and zero virtual dollars when they guessed incorrectly. In designs 2 and 3, subjects received two virtual dollars when they guessed correctly and one virtual dollar when they guessed

**Table 1. Experimental design.**

| Design | Image Probability | Payoff | Utility Maximizing Behavior | Evolutionarily Dominant Behavior [50] |
|:---:|:---|:---:|:---:|:---:|
| **1** | $\mathbb{P}(\text{Angelina}) = 0.7$ <br> $\mathbb{P}(\text{Brad}) = 0.3$ | Correct: v\$2 <br> Incorrect: v\$0 | Always Guess Angelina | $f^* = \mathbb{P}(\text{Guess Angelina}) = 0.7$ |
| **2** | $\mathbb{P}(\text{Angelina}) = 0.7$ <br> $\mathbb{P}(\text{Brad}) = 0.3$ | Correct: v\$2 <br> Incorrect: v\$1 | Always Guess Angelina | $f^* = \mathbb{P}(\text{Guess Angelina}) = 1.0$ |
| **3** | $\mathbb{P}(\text{Brad}) = 0.7$ <br> $\mathbb{P}(\text{Angelina}) = 0.3$ | Correct: v\$2 <br> Incorrect: v\$1 | Always Guess Brad | $f^* = \mathbb{P}(\text{Guess Brad}) = 1.0$ |
| **4** | $\mathbb{P}(\text{Brad}) = 0.7$ <br> $\mathbb{P}(\text{Angelina}) = 0.3$ | Correct: v\$2 <br> Incorrect: v\$0 | Always Guess Brad | $f^* = \mathbb{P}(\text{Guess Brad}) = 0.7$ |

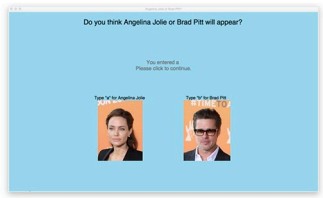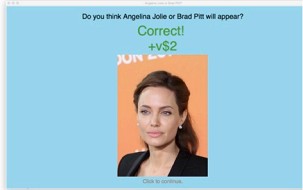

**Fig 2. Screenshots of the experimental program.** The left image shows the screen before the user submits her guess, and the right image shows the screen displaying the result of a correct guess. The pictures of Angelina Jolie and Brad Pitt are republished from *Foreign, Commonwealth & Development Office* under a CC BY license, with permission through creativecommons.org, original copyright 2014. We use different images of Angelina Jolie and Brad Pitt in the study, and the screenshots here are therefore for illustrative purposes only.

incorrectly. These designs correspond to four different evolutionarily dominant behaviors in Fig 1 (see also Brennan and Lo [50]), as shown in the last column of Table 1. Designs 1 and 4 are meant to yield randomized behavior according to theory, while designs 2 and 3 are meant to yield deterministic behavior. In terms of parameters in Fig 1 using $p = 0.7$, Design 1 corresponds to $r_1 = \infty$ and $r_2 = 0$, which yields the dominant behavior $f^* = 0.7$; Design 2 corresponds to $r_1 = 2$ and $r_2 = \frac{1}{2}$, which yields the dominant behavior $f^* = 1$; Design 3 corresponds to $r_1 = \frac{1}{2}$ and $r_2 = 2$, which yields the dominant behavior $f^* = 0$; Design 4 corresponds to $r_1 = 0$ and $r_2 = \infty$, which yields the dominant behavior $f^* = 0.3$.

Fig 3 shows the trial-by-trial outcome of two representative subjects. The subject in Fig 3a had a mix of guesses of Angelina Jolie and Brad Pitt throughout the duration of the experiment. We refer to these subjects as "randomizers.' On the other hand, the subject in Fig 3b guessed Angelina Jolie for almost the entire duration of the experiment; her only Brad Pitt guesses occurred within her first few attempts, when it is likely that she was still determining the pattern. We refer to these subjects as "maximizers."

We provided an extensive explanation of the game for all participants and answered any questions they had before the start. In order to ensure the subjects' comprehension of the mechanics of the game, we conducted 50 iterations of a control game for each subject, either before or after the real experiment. The control game still showed the two images of Angelina Jolie and Brad Pitt at random, but the subjects were explicitly told that they only would earn a reward by always guessing one of the images, randomized between Angelina Jolie and Brad Pitt, but fixed for any one subject. Subjects who understood the payoff structure should always guess the subject with the reward. This was a trivial game, used to test the participant's understanding of the main lottery game.

The majority of our subjects passed the control and demonstrated a clear understanding of the payoff. However, a total of 7 subjects guessed the image with zero payoff in the control game more than 40% of the time, and we concluded that they were not paying proper attention to the task, and discarded their data in our subsequent analysis. This left us with valid data from a total of 75 subjects.

## 3 Results

### 3.1 Summary statistics

After participating in the experiment, all participants completed a personal information survey form. Table 2 contains summary statistics from these surveys, including the subjects' personal and socioeconomic characteristics, as well as their game responses. Although our sample is fairly balanced in terms of gender, age, working status, income, wealth, and investment experience, it is skewed toward single (77.3%) subjects and those without children (86.7%). In

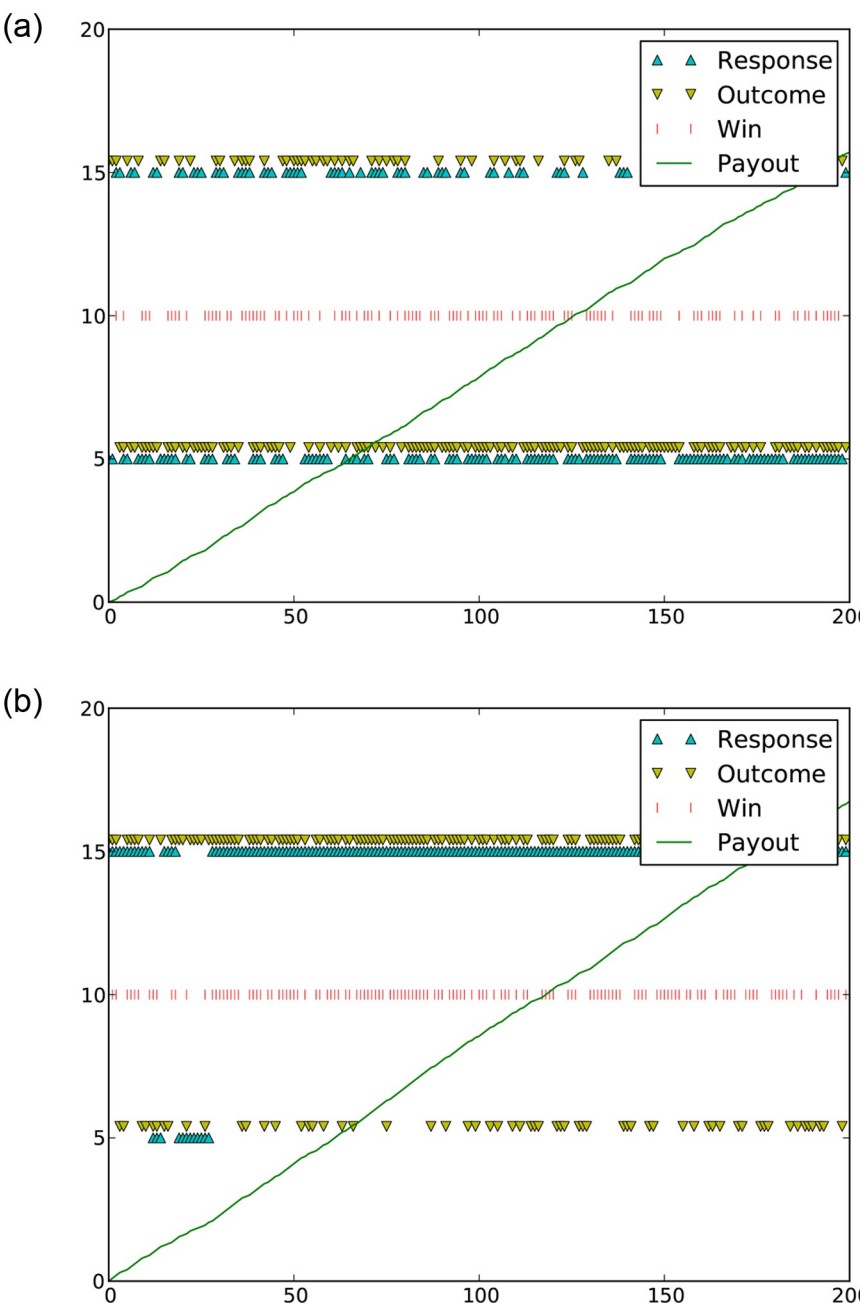

**Fig 3. Experimental outcomes for a representative randomizer (3a) and a representative maximizer (3b).** The highest row of triangles displays the randomly generated appearances of Angelina Jolie for each trial. The second row of triangles displays the instances when the subject's response was Angelina Jolie. The bottom two rows of triangles represent the same information for Brad Pitt appearances and Brad Pitt responses. The middle row of red ticks represents trials that the subject guessed correctly. The diagonal line shows the cumulative payout to the subject over time.

addition, 64% of our subjects have reported taking some probability and statistics classes, an unsurprising finding, given that the experiment took place at MIT.

Subjects each received $5 in base pay for showing up, and $0.05 for each virtual dollar they earned. Total dollar earnings ranged from $14.80 to $22.20. Table 2 also reports the total

**Table 2. Participant demographics and summary statistics.**

| Variable | Distribution (n = 75 subjects) | | | | |
|---|---|---|---|---|---|
| | **Personal Characteristics** | | | | |
| **Gender** | Male 53.3% | Female 45.3% | | | |
| **Has Children** | Yes 12.0% | No 86.7% | | | |
| **Age** | ≤23 33.3% | [24, 46] 34.7% | >46 30.7% | | |
| **Marital Status** | Single 77.3% | Partnered 9.3% | Married 8.0% | Other 4.0% | |
| | **Socioeconomic Characteristics** | | | | |
| **Taken Probability & Statistics Class** | Yes 64.0% | No 34.7% | | | |
| **Gambling Experience** | Yes 33.3% | No 65.3% | | | |
| **Housing Status** | Rent 88.0% | Own 10.7% | | | |
| **Working Status** | Student 40.0% | Currently Working 38.7% | Unemployed & Other 20.0% | | |
| **Annual Family Income** | <$25,000 30.7% | $25,000-$50,000 24.0% | $50,001-$100,000 28.0% | >$100,000 16.0% | |
| **Total Financial Assets** | <$25,000 62.7% | $25,000-$50,000 9.3% | $50,001-$100,000 14.7% | >$100,000 12.0% | |
| **Investment Horizon** | I don't invest 52.0% | Less than 1 year 9.3% | Over 1 year 37.3% | | |
| | **Game-related Questions** | | | | |
| **Found Pattern in Game (Self-reported)** | Yes 44.0% | No 48.0% | I don't remember 6.7% | | |
| **Had Strategy in Game (Self-reported)** | Yes 74.7% | No 22.7% | I don't remember 1.3% | | |
| | **Game Outcome Percentiles** | | | | |
| **Correct Guesses (Out of 200 Total)** | Min 98 | 5% 108 | 50% 129 | 95% 143 | Max% 154 |
| **Total Earnings ($) (Including $5 Base)** | Min 14.8 | 5% 15.8 | 50% 20.3 | 95% 22.0 | Max% 22.2 |

number of correct guesses for all subjects. The best performer guessed 154 (77%) trials correctly, while the worst performer only guessed 98 (49%) trials correctly. The median subject guessed 129 (64.5%) out of the 200 trials correctly, slightly less than the expected number of correct guesses for a perfect maximizer, who would always guess the dominant image.

The post-game survey also asked participants about their perceptions of the binary choice game. 44% of our subjects reported that they found a pattern in the game. It is clear that many participants were looking for patterns throughout the game, despite its completely random nature. This is consistent with the "representativeness heuristic" first documented by Tversky and Kahneman [61, 62]. We include quotes from two representative subjects.

"I kept losing count, but clearly the ratio of appearance of Jolie's picture to Pitts's kept going up until it was something 7:1, then it went down (not always in increments of one, I think) until it was 1:1, and then it went back up again."

"70% Angelina. If we picked her too many times, Brad was introduced as a counter-pick."

In addition, 74.7% of the subjects reported that they had a specific strategy in the game. By reading the post-study surveys, we realized that our subjects exhibited a wide range of heterogeneous strategies for the game. Here we show a few representative quotes from the two extremes of these strategies, where some subjects indicated clearly that they were always choosing one image:

"Always pick Angie."

"Choosing Brad Pitt all the time. His image appeared more frequently and even if the probability was 50% it would not have mattered who I choose, so why not choose him all the time. Also minimizes thinking effort and time to click."

Other subjects seemed to engage in more complicated strategies:

"Chose Brad Pitt the majority of the time—if Brad Pitt appeared at least 6 times in a row, chose Angelina Jolie."

"'. . .I was switching between one and another until I noticed some sort of pattern and then I favored Angelina Jolie's picture for the higher number and Brad Pitt for the lower number in the pattern of 5-1-3-1-2."

These self-reported strategies are also reflected in the wide heterogeneity in behavior when we analyze participant choices.

## 3.2 A model for individual behavior

Brennan and Lo [50] predict that subjects assigned to designs 1 and 4 of our binary choice game will randomize their behavior. We refer to them as "theoretical randomizers." On the other hand, subjects assigned to designs 2 and 3 are predicted to choose the dominant image deterministically, and we refer to them as "theoretical maximizers" (see Table 1). In this section, we study whether theoretical randomizers indeed randomize more often than theoretical maximizers.

We first describe a simple model of individual behavior. Define $D$ to be the dominant option in the game. In our experiment, $D$ represents Angelina Jolie in designs 1 and 2, and Brad Pitt in designs 3 and 4.

Each individual $i$ chooses the dominant option $D$ with probability $f$, where $f$ represents the individual's (unobserved) behavior. In other words, the individual's decision in each trial is generated by a Bernoulli random variable:

$$y_t = \begin{cases} 1, & \text{with probability } f, \\ 0, & \text{with probability } 1-f, \end{cases} \tag{2}$$

where $y_t = 1$ represents choosing the dominant option $D$, and $t = 1, \ldots, 200$. Suppose in $T$ trials, an individual chooses the dominant option

$$N := \sum_{t=1}^{T} y_t \tag{3}$$

times. From observed data $T$ and $N$, our goal is to estimate and understand the factors which

determine individual behavior $f$ in different payoff structures. The sample average proportion

$$\hat{f} := N/T \qquad (4)$$

is the obvious choice as the point estimate of behavior $f$.

If an individual's decisions are independent over time, it follows from (2) and (3) that $N \sim Binomial(T, f)$, and $\hat{f}$ is approximately normally distributed with mean $f$ and variance $f(1-f)/T$. More generally, if an individuals' decisions are not independent over time, $\hat{f}$ still has mean $f$, but its variance may be different. In Section 3.4, we estimate whether individual decisions are independent, and in Section 3.5 we discuss its implications for the variance of $\hat{f}$ and the hypothesis tests we carry out.

### 3.3 Initial learning

During the experiment, subjects required a number of trials to estimate the frequency of each image. This means that their first few guesses tended to show unstable behavior. To account for this, we divided each individual's total number of trials into eight consecutive segments of 25 trials each, and estimated the aggregate behavior $f$ for each segment across individuals within the same trial design. Individual behavior was too noisy for successful functional estimates over the initial trials, so we used the aggregate pattern across individuals to better understand the speed of participant learning.

Fig 4 shows the estimated aggregate behavior for theoretical maximizers (designs 2 and 3, $f^* = 0.7$) and theoretical randomizers (designs 1 and 4, $f^* = 1.0$), segmented into eight consecutive batches. We used the sample average proportions $\hat{f}$ in (4) as the point estimate of behavior $f$, and the normal approximation for binomial distributions to estimate its confidence interval: $\hat{f} \sim N(f, f(1-f)/T)$. More specifically, for a given confidence level $1 - \alpha$ (e.g., $\alpha = 0.05$, or 95% confidence), the $(1 - \alpha)$-confidence interval is given by:

$$\left( \hat{f} - z\sqrt{\frac{\hat{f}(1-\hat{f})}{T}}, \hat{f} + z\sqrt{\frac{\hat{f}(1-\hat{f})}{T}} \right)$$

where $z$ is the $1 - \frac{\alpha}{2}$ quantile of a standard normal distribution corresponding to the target error rate $\alpha$. For a 95% confidence level, the error $\alpha = 1 - 0.95 = 0.05$, so $1 - \frac{\alpha}{2} = 0.975$ and $z = 1.96$. There are other approximations for confidence intervals of binomial random

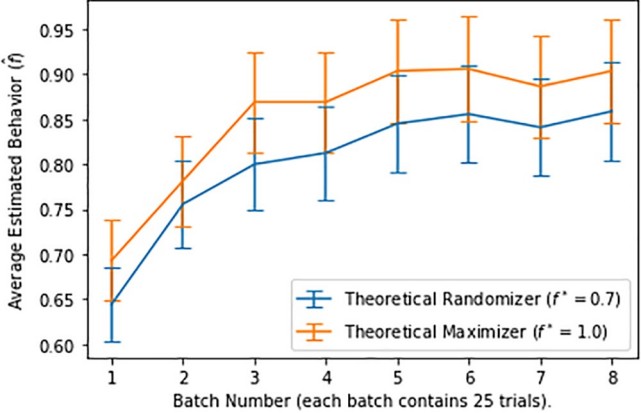

**Fig 4. Estimated aggregate behavior for batches of 25-trial segments.**

variables, such as the Wilson score interval and the Jeffreys interval [63, 64]. We use the normal approximation for simplicity.

We see that the estimated behavior in the first two batches (the first 50 trials) are lower than in the remaining batches, starting to stabilize around batch 3. For consistency across all individuals, we consider the responses starting from trial 51 as stable, and do not include the first 50 trials in our subsequent analysis. Our main conclusions in this paper are not sensitive to this choice.

### 3.4 Decision autocorrelation

In general, the distribution of our estimated behavior $\hat{f}$ depends on the covariance structure of an individuals' decisions over time: $\{y_t\}_{t=1}^{T}$. For individuals' stable trials (trial 51 to 200), we do find evidence for autocorrelation in individual decision making. In fact, applying the Ljung–Box $Q$ test to each individual's sequence of choices yields 38 $p$-values that are smaller than 0.05 (34 after the Benjamini–Hochberg procedure to control for the False Discovery Rate in multiple testing), out of 75 subjects. This indicates that roughly half of the subjects show significant autocorrelation.

To incorporate correlation into the variance estimate of $\hat{f}$ in (4), we compute an autocorrelation function based on all individuals' stable trials. Specifically, the lag-$l$ autocorrelation, ACF($l$), is estimated using the sample correlation of the following pairs:

$$\{(y_t^{(i)}, y_{t+l}^{(i)})\}_{t=51,\cdots,T-1, \text{ and } i=1,\cdots,n} \tag{5}$$

where the superscripts ($i$) denotes the $i$-th subject, which we normally omit for simplicity elsewhere in the paper.

Fig 5 shows the autocorrelation function up to lag 120. The autocorrelation is 32.4% at lag-1 and quickly stablizes around 8%-12%. We also pool together autocorrelations at all lags to yields an estimate of equicorrelation of 10.8%, which we use in the next section to test the hypothesis whether the behavior for theoretical maximizers is different from the behavior for theoretical randomizers.

### 3.5 Probability matching

The evolutionary model of probability matching [50] predicts that the behavior in design 2 and 3 should be $f_{maximizer} = 1.0$, while the behavior in design 1 and 4 should be $f_{randomizer} = 0.7$. This of course is under the hypothetical condition that the only determinant of an individual's

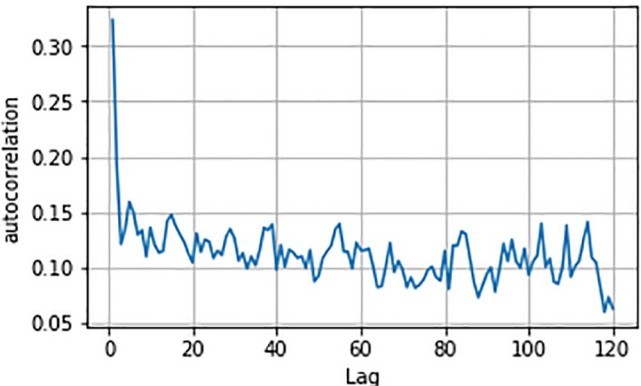

**Fig 5. Autocorrelation function estimated from all individuals' stable trials.**

reproductive success is the payoff from the game. This is obviously an extreme simplification of reality. Nonetheless, the model still provides the important insight that the presence of probability matching, and the degree at which individuals engage in probability matching, are determined by the environment, which we can specifically test through our experiment.

In particular, we are able to test the hypothesis that:

$$H_0 : f_{maximizer} \leq f_{randomizer}, \quad H_a : f_{maximizer} > f_{randomizer} \tag{6}$$

as predicted by Brennan and Lo [50], where $f_{maximizer}$ is the behavior of individuals in designs 2 and 3, and $f_{randomizer}$ is the behavior of individuals in designs 1 and 4. As specified in (2), we observe repeated individual decisions that are, in our model, determined by the unobserved behavior $f_{maximizer}$ and $f_{randomizer}$. We can pool together data from all theoretical maximizers and compare with data from all theoretical randomizers. This is a standard two sample proportion test, except that decisions for the same individual might be correlated, as shown in Section 3.4.

Given a particular individual, we use vector $\mathbf{y} := (y_1, \cdots, y_T)'$ to denote her sequence of $T$ random Bernoulli trials. For simplicity and analytical tractability, we assume the sequence has equicorrelation of $\rho$ (estimated as 10.8% in our dataset). In other words, the covariance matrix of $\mathbf{y}$ is given by

$$\text{Cov}(\mathbf{y}) = \text{Var}(y_1) \cdot \text{Corr}(\mathbf{y}) = f(1-f) \cdot \begin{pmatrix} 1 & \rho & \cdots & \rho \\ \rho & 1 & \cdots & \rho \\ \cdots & & & \\ \rho & \rho & \cdots & 1 \end{pmatrix}, \tag{7}$$

where the first equation follows from the fact that $y_1, \cdots, y_T$ are identically distributed as specified in (2).

Therefore, the variance of the estimated behavior $\hat{f}$ for individual $i$, as defined in (4), is given by

$$\begin{aligned} \text{Var}(\hat{f}^{(i)}) &= \text{Var}\left(\frac{\sum_{t=1}^T y_t}{T}\right) = \frac{\text{Var}(\mathbf{1}'\mathbf{y})}{T^2} = \frac{\mathbf{1}'\text{Cov}(\mathbf{y})\mathbf{1}}{T^2} \\ &= \frac{f(1-f)T(1+(T-1)\rho)}{T^2} \\ &= \frac{f(1-f)}{T} \cdot (1+(T-1)\rho) \end{aligned} \tag{8}$$

where $\mathbf{1}$ is the unit vector of all 1's, and we have omitted the superscript $(i)$ in our derivation for notational simplicity.

Note that the first term in (8), $\frac{f(1-f)}{T}$, is simply the variance of $\hat{f}^{(i)}$ if individual decisions are independent Bernoulli random variables. Therefore, the second term in (8), $(1+(T-1)\rho)$, can be treated as an adjustment factor of $\hat{f}^{(i)}$'s variance when individual decisions are correlated.

For a set of $n$ independent individuals with $T$ trials each, the overall estimated behavior for them is simply the average estimated behavior of each individual. Therefore the variance for their overall behavior is:

$$\text{Var}\left(\hat{f}\right) = \text{Var}\left(\frac{\sum_{i=1}^n \hat{f}^{(i)}}{n}\right) = \frac{\text{Var}(\hat{f}^{(1)})}{n} = \frac{f(1-f)}{nT} \cdot (1+(T-1)\rho).$$

As a result, for an (unpaired) two sample proportion test between two groups of subjects with $n_1$ and $n_2$ individuals respectively, we have the test statistic:

$$z = \frac{(\hat{f}_1 - \hat{f}_2) - 0}{\sqrt{f^*(1-f^*)\left(\frac{1}{n_1 T} + \frac{1}{n_2 T}\right)}} \cdot \frac{1}{\sqrt{1 + (T-1)\rho}} \tag{9}$$

where $\hat{f}_1, \hat{f}_2$, and $f^*$ are the average behavior for individuals from group 1, group 2, and all pooled together, and they do not depend on $\mathbf{y}$'s covariance structure. The first term in (9) is simply the standard $z$-score for the two sample proportion test, and the second term in (9) can be treated as the adjustment factor for correlation, which in our case is:

$$\frac{1}{\sqrt{1 + (T-1)\rho}} = \frac{1}{\sqrt{1 + (150 - 1) \cdot 10.8\%}} \approx 0.24$$

for stable trials.

With this correlation adjustment, the null hypothesis in (6) is rejected with a $z$-statistic of 2.014 (or a $p$-value of 0.022), providing evidence for a difference in behavior between theoretical maximizers and theoretical randomizers. As predicted, when facing different environments (i.e., different payoffs in the experiment), theoretical randomizers indeed randomize more often than theoretical maximizers. Subjects responded differently by adapting to the environment and showing different stable behaviors.

After the game, we asked subjects in a survey whether they employed a specific strategy, and 74.7% of the subjects reported "Yes". We perform the same hypothesis test (6) for subjects who reported "Yes" and those who reported "No" separately. We find that the effect holds strongly for individuals who reported that they used a specific strategy (adjusted $z$-statistic of 2.489, adjusted $p$-value of 0.006), but not for those who did not (adjusted $z$-statistic of $-0.058$, adjusted $p$-value of 0.523). This serves as another robustness check that the effect is driven by intentional behavior on the part of the subjects, and is not purely noise. This also provides empirical evidence for theories that attempt to explain probability matching through pattern seeking [55] and searching for causal relationships [45].

In principle, one can also perform the same test for different slices of the subjects across demographic, socioeconomic, and game-specific dimensions shown in Table 2. This helps to build intuition on whether the same effect holds true universally, and what types of individuals have stronger effects. However we acknowledge that the power of our study is limited due to the sample of 75 subjects, particularly after multiple-testing adjustment, and we leave this to a future study.

### 3.6 Individual differences

To jointly study individual differences in decision-making with the variables considered in Table 2, we consider a logistic regression model at the level of each guess by the individual subject. Specifically, for individual $i$, at the $t$-th trial in the game,

$$y_t^{(i)} \sim Bernoulli(f^{(i)})$$

where $y_t^{(i)} = 1$ is a binary random variable that represents whether individual $i$ chooses the dominant option at trial $t$, similar to the formulation in Eq (2). Individual $i$'s behavior—the

probability of choosing the dominant option—is modeled by:

$$
\begin{aligned}
f^{(i)} &= \mathbb{P}(y_t^{(i)} = 1) \\
&= Logistic(IsTheoryMaximizer_i + IsMale_i + HasChild_i + AgeBucket_i \\
&\quad + HasProbClassExp_i + HasGamblingExp_i + HasInvestExperience_i \\
&\quad + IsStudent_i + IsOwn_i + IncomeBucket_i + TotalAssetBucket_i \\
&\quad + HasPattern_i)
\end{aligned}
\tag{10}
$$

where $Logistic(x) = (1 + \exp(-x))^{-1}$.

We have seen in Section 3.4 that individual decisions are correlated over time. Therefore, the errors for regression (10) may be autocorrelated. We group trials from the same individual together, and order their decisions chronologically. In particular, the response variable is organized as:

$$
(\underbrace{y_1^{(1)}, y_2^{(1)}, \cdots, y_T^{(1)}}_{\text{1st subjects } T \text{ trials}}, \underbrace{y_1^{(2)}, y_2^{(2)}, \cdots, y_T^{(2)}}_{\text{2nd subjects } T \text{ trials}}, \cdots, \underbrace{y_1^{(n)}, y_2^{(n)}, \cdots, y_T^{(n)}}_{n\text{-th subjects } T \text{ trials}})',
$$

and we apply the Newey-West heteroskedasticity and autocorrelation consistent (HAC) estimator for the variance of the coefficients in the following results. We adopt Newey and West's [65] suggestion to choose the truncation parameter to be the integer part of $4(nT/100)^{2/9}$, which is 11 in our case. This indeed increases the variance estimate of our coefficients compared with the case of independent errors, and our results are not materially different with respect to the choice of the truncation parameter. In fact, we have tried an estimation with truncation parameter to be 150, the number of total valid decisions for one individual. The main variable *IsTheoryMaximizer* remains statistically significant at 5%.

Table 3 summarizes the independent variables in Eq (10). These variables correspond to the collected personal information of the subjects (see Table 2), categorized to make them proper binary or ordinal variables. We have dropped several variables that are highly collinear with the covariates in (10). The $p$-value of the log-likelihood ratio test of the full model is $4 \times 10^{-54}$, implying a high degree of significance.

**Table 3. Variables and coefficients for logistic regression.**

| Variable | Type | Coefficient | HAC Standard Error | z Statistic |
|---|---|---|---|---|
| IsTheoryMaximizer | Dummy | 0.4180 | 0.106 | **3.959**\*\*\* |
| IsMale | Dummy | 0.1906 | 0.100 | 1.901 |
| HasChild | Dummy | -0.5497 | 0.126 | **-4.378**\*\*\* |
| AgeBucket | Ordinal | -0.0747 | 0.078 | -0.962 |
| HasProbClassExp | Dummy | -0.2832 | 0.116 | **-2.445**\* |
| HasGamblingExp | Dummy | 0.0120 | 0.106 | 0.113 |
| HasInvestExperience | Dummy | 0.0393 | 0.108 | 0.365 |
| IsStudent | Dummy | -0.1001 | 0.128 | -0.78 |
| IsOwn | Dummy | -0.2603 | 0.206 | -1.266 |
| IncomeBucket | Ordinal | 0.0246 | 0.038 | 0.642 |
| TotalAssetBucket | Ordinal | 0.0965 | 0.048 | **2.022**\* |
| HasPattern | Dummy | -0.5497 | 0.101 | **-5.467**\*\*\* |
| Intercept | | 2.1051 | 0.160 | **13.126**\*\*\* |

Statistics in **bold** are significant at the 0.1% (\*\*\*), 1% (\*\*), or 5% (\*) level.

The first variable, IsTheoryMaximizer, encodes whether the individual is a theoretical maximizer, i.e., placed in design 2 or 3. The effect is both positive and strongly significant after controlling for all other socioeconomic and game-related variables, implying that theoretical maximizers tend to choose the dominant option more often. This is consistent with our analysis in Section 3.5.

Continuing down the table, a significantly positive coefficient corresponds to a dimension along which individuals tend to be a maximizer, i.e., they randomize less often. Our results show that subjects with more financial assets tend to be maximizers more often than subjects with fewer financial assets.

A significantly negative coefficient, on the other hand, corresponds to a dimension along which individuals tend to be a randomizer, i.e., they switch between two options more often. Our results show that subjects with children tend to be randomizers.

In addition, subjects who have taken probability and statistics classes and who self-reported finding a pattern in the game tend to randomize more. This is contrary to our prior expectation that those who understand probability might be more likely to always choose the dominant option, the economically maximizing behavior. In fact, they behave in the exactly opposite manner, perhaps in an attempt to "beat the game." This is consistent with our observations of subject narratives in the post-study survey in Section 3.1.

## 4 Discussion

We have used a simple lottery game to test the occurrence of probability matching behavior in financial decision-making. Our study specifically uses a choice between two images rather than more financially-related tasks (such as guessing the outcome of asset-price movements) in the hopes of triggering a more primitive form of decision-making in our subjects, because financially-related tasks may be more likely to trigger economically optimizing behaviors. Despite a large degree of heterogeneity among individual behaviors, we find that different levels of probability matching occur in different environments of payoff structures. Specifically, individuals in environments with less balanced payoffs between the two random outcomes (design 1 and 4 in Table 1) show a greater degree of randomized behavior, consistent with the behavioral predictions of Brennan and Lo [50].

On the other hand, we did not observe individuals behave in exactly the way the simple theoretical model would suggest ($f^* = 0.7$ for design 1 and 4; $f^* = 1.0$ for design 2 and 4 in Table 1), most likely because the real effects on the reproductive success of each individual are unlikely to be affected by the payoffs provided in the game, but instead are affected by a number of heterogeneous socioeconomic factors such as income, wealth, and other variables. It is difficult, and perhaps impossible, to specify a single model that predicts behavior for every participant under all circumstances. However, we have indeed observed that at the population level, the effect that theoretical randomizers (design 1 and 4) randomize more than theoretical maximizers (design 2 and 3) is real and robust, after controlling for a wide range of such personal and socioeconomic variables.

We find significant evidence of different levels of probability matching behavior when we alter the payoff structure in the game. However, our study is also constrained by the limited magnitude of payoffs (around $20) offered to subjects, as well as the limitations of our sample size (82 subjects). It is therefore difficult to perform additional statistical tests on finer slices of the data to study whether the same behavioral mechanism applies to different demographic categories. It is also difficult to conclude that any demographic dimension that appears neutral to individual decision making would remain neutral in a larger-scale study. It remains a question for future research to confirm whether our experimental conclusions carry over across

demographic groups, to other financial and non-financial contexts, and at different magnitudes of payoffs.

In addition to testing the evolutionary model of Brennan and Lo [50], our experimental results suggest that it is valuable to derive behavioral predictions and implications through an evolutionary lens. Traditional utility-based theories would yield the same maximizing behavior for all four designs in our experiment. Yet we find evidence for differences in reality, and the evolutionary framework offers a potential explanation and prediction for such behaviors: the environment matters.

More generally, financial markets—a collection of individual decision makers—can also be studied using the same principles, leading to the Adaptive Markets Hypothesis [66, 67] and its many empirical implications [68–70]. In the same way that micro-level individual decision making can be better understood through an evolutionary lens, markets and societies at the system-wide and macroscopic level can also benefit an adaptive perspective.

## Acknowledgments

The authors thank the MIT Behavioral Research Laboratory for hosting the experiment, and Jennifer L. Walker, Jayna Cummings, and Allison McDonough for their assistance in the experiments. Insightful comments and discussions from Jason Anthony Aimone (editor), Kim Fairley (reviewer), Taibo Li and Alex Huang are also greatly appreciated.

## Author Contributions

**Conceptualization:** Andrew W. Lo, Katherine P. Marlowe, Ruixun Zhang.

**Data curation:** Andrew W. Lo, Katherine P. Marlowe, Ruixun Zhang.

**Formal analysis:** Andrew W. Lo, Katherine P. Marlowe, Ruixun Zhang.

**Investigation:** Andrew W. Lo, Katherine P. Marlowe, Ruixun Zhang.

**Methodology:** Andrew W. Lo, Katherine P. Marlowe, Ruixun Zhang.

**Supervision:** Andrew W. Lo.

**Validation:** Andrew W. Lo, Katherine P. Marlowe, Ruixun Zhang.

**Writing – original draft:** Andrew W. Lo, Katherine P. Marlowe, Ruixun Zhang.

**Writing – review & editing:** Andrew W. Lo, Katherine P. Marlowe, Ruixun Zhang.

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
