## [Decision Letter · Decision Letter 0]

22 Feb 2021

PONE-D-20-40629

To maximize or randomize? An experimental study of probability matching in financial decision making

PLOS ONE

Dear Dr. Zhang,

Thank you for submitting your manuscript to PLOS ONE. After careful consideration, we feel that it has merit but does not fully meet PLOS ONE’s publication criteria as it currently stands. Therefore, we invite you to submit a revised version of the manuscript that addresses the points raised during the review process.

The decision to request a revision comes after consultation with a qualified reviewer who raises important points about the paper that need to be addressed. While I ask you to look at and respond to each of the points, of particular importance is ensuring that the data analysis is correctly completed. Note for instance the lack of description in the text about how repeat observations from individual subjects is treated in the regression analysis.  I gave the paper a read myself and have some points I would like you to address as well. Several of these points are dealing with the manner in which the paper is discussed and where confusion may arise with readers. There are also several statistical points of concern to be addressed.

P1/2: Discussion of rationality makes it sound like the factors identified that lead to differences to the things discussed in the first paragraph are “irrational”, which the lines of work discussing those things do not always indicate.  Loss aversion for instance is not necessarily “irrational” even though it goes against what could be called “traditional” theoretical models of rationality.The probability matching example of coin flips does not do what it is trying to do. I’d suggest rewording or providing a more detailed example.  The intro needs a good example for those who are not familiar with the concept.You say “It is interesting to note that the t-statistic is higher for those subjects who reported “Yes” than for those who reported “No”. This implies that subjects who believed there was a pattern were adapting to the environment of the game more strongly”.  Please directly test for this difference.Table 3:  It is not clear how you are setting up these tests from the text. It would seem reasonable to be identifying the “f” for each participant as the probability they choose the dominant option, and thus there would be 1 data point per individual. You then would make comparisons between your four treatments. However, you say “The number of samples in the t-test equals the number of subjects multiplied by 150 stable trials.”  Are you saying here that you are treating each subject as 150 different independent observations (which would be inappropriate?)  This is confusing and muddles the whole results section.Discussion:  The conclusion is written strangely and leads to the opposite conclusions as the paper. “Specifically, individuals in environments with more balanced payoffs between the two random outcomes show a greater degree of randomized behavior, consistent with the behavioral predictions in Brennan and Lo [44]” Your paper sets up behavior in treatments 1 and 4 (That pay for correct and do not pay for incorrect, thus very unbalanced payments) are more likely to result in decision randomization than treatments 2 and 3 (where people get paid relatively more equally for correct and incorrect answers.) This is opposite of the quote from the conclusion. Why? Your regression show analysis shows that the variable “Theory Maximizer” which you say is 1 if in treatments 2 or 3, has a positive coefficient which you say is interpreted as meaning they choose the dominant option more often, which would seem to be the opposite of engaging in more randomized behavior.  The conclusion continues to say that the data did not support the table 1 theory.  That data did appear to support the Evolutionarily Dominate behavior theoretical outcome.  Is there something I am missing or missed in this explanation?

Finally, I head some comments back from a second reviewer who recognized late in the process a conflict of interest and withdrew from the formal process. I will pass on a few thoughts for your revision from that reviewer though. I summarize here:

*p1, ln 13-14 should be rephrased as as overconfidence and loss aversion are less "behaviors" and more psychological traits

*p8, elaborate and clarify the position that pattern seeking might be a form of randomization.

*p6, quote starting "the experiment we describe...for those environments." you may want to emphasize this in the introduction.

We look forward to receiving your revised manuscript.

Kind regards,

Jason Anthony Aimone

Academic Editor

PLOS ONE

Journal Requirements:

Reviewers' comments:

Reviewer's Responses to Questions

**Comments to the Author**

1. Is the manuscript technically sound, and do the data support the conclusions?

Reviewer #1: Partly

2. Has the statistical analysis been performed appropriately and rigorously? 

Reviewer #1: No

3. Have the authors made all data underlying the findings in their manuscript fully available?

Reviewer #1: No

4. Is the manuscript presented in an intelligible fashion and written in standard English?

Reviewer #1: Yes

5. Review Comments to the Author

Reviewer #1: To maximize or randomize? An experimental study of probability matching in financial

decision making

By Andrew W Lo, Katherine P Marlowe, Ruixun Zhang

Reviewed on Feb 11, 2021

Summary

This study aims to experimentally test the model predictions by Brennan and Lo (2011) on probability matching. By probability matching the authors refer to a behavioural tendency to predict the outcomes of an independent randomized event to match it underlying probability distribution. In other words, participants do not pick one outcome based on maximum utility, but pick various outcomes which correspond to the underlying probability with which these outcomes occur.

Their experimental design was tested on 82 participants in a behavioural laboratory setup and consisted of 200 trials of a binary choice decision. This binary choice involved guessing if an image of Brad Pitt or Angelina Jolie would appear and with every good guess participants would accrue experimental tokens. By varying the payoff structure and the probability with which one of the two images would appear, four experimental conditions were created. Based on the evolutionary dominant strategies as laid out by Brannon and Lo (2011), experimental results indeed showed a behavioural difference between theoretical maximizers and randomizers.

Major concerns

1. Aims of this paper

The goal of this paper is to experimentally test the evolutionary model by Brannan and Lo. This aim should be made much clearer than at the end of the Introduction, where it is written now. Although the authors describe this model in some detail in 2.1, the specific environmental conditions (which leads the authors to design their payoff structure) that underlies probability matching according to the authors are not well described. As this model forms the backbone of the experimental predictions, it should be described in an intuitive clear manner.

2. Power analysis

Did the authors perform a power analysis? As many control variables are included, I wonder if there is enough power to test the numerous relationships as laid out in the results section with only 82 participants.

3. Separate t-tests

It is weird to run separate t-tests for each and every socio-demographic variable. You would need to perform a correction for multiple hypothesis testing to begin with, but also I do not see the purpose of running these t-tests. I would just stick to your logistical regression. With regard to your logistical regression, how do you control for the repeated trials at the subject level, as this probably violates independence of individual data points? This is unclear and needs to be explained better.

4. Discussion

Your discussion is a summary of your results and does not relate your findings to any other study, does not discuss any limitations of your study in much detail, or addresses the relevance and practical implications of your discussion. In short, the discussion does not read as a discussion.

Minor concerns

5. Probability matching is also a well-known term to label a technique to elicit ambiguity preferences. I would suggest you make it clear very early on what probability matching entails in your study.

6. You explain probability matching as randomizing behaviour, but also refer to randomizing and probability as two phenomena. It is unclear if these labels are synonyms or not.

7. What do we learn from the neuroimaging and tDCS studies regarding probability matching in your literature review? These insights feel disconnected from the rest of the literature review. Also, I think many have no clue what a tDSC study entails, so either explain better and embed within your framework, or leave it out.

8. Why did you pick such valanced images as Brad and Angelina and did not go for a neutral binary choice? Especially, since your aim is to ‘use a choice between two images rather than a more financially-related tasks (such as guessing the outcome of asset-price movements) in the hopes of triggering a more primitive form of decision-making in our subjects’. (r. 425-427). What is primitive about choosing between Brad and Angelina?

6. PLOS authors have the option to publish the peer review history of their article (what does this mean?). If published, this will include your full peer review and any attached files.

Reviewer #1: **Yes: **Kim Fairley

---

## [Author Response · Author response to Decision Letter 0]

6 Apr 2021

We thank both the editor and the reviewer for their very helpful comments on our work—they have greatly improved the manuscript. The revised manuscript incorporates all of these comments and suggestions and point-by-point responses are provided. We have also provided a revised manuscript with track changes.

---

## [Decision Letter · Decision Letter 1]

18 May 2021

To maximize or randomize? An experimental study of probability matching in financial decision making

PONE-D-20-40629R1

Dear Dr. Zhang,

We’re pleased to inform you that your manuscript has been judged scientifically suitable for publication and will be formally accepted for publication once it meets all outstanding technical requirements.

Kind regards,

Jason Anthony Aimone

Academic Editor

PLOS ONE

Additional Editor Comments (optional):

Reviewers' comments:

Reviewer's Responses to Questions

**Comments to the Author**

1. If the authors have adequately addressed your comments raised in a previous round of review and you feel that this manuscript is now acceptable for publication, you may indicate that here to bypass the “Comments to the Author” section, enter your conflict of interest statement in the “Confidential to Editor” section, and submit your "Accept" recommendation.

Reviewer #1: All comments have been addressed

2. Is the manuscript technically sound, and do the data support the conclusions?

Reviewer #1: Yes

3. Has the statistical analysis been performed appropriately and rigorously? 

Reviewer #1: Yes

4. Have the authors made all data underlying the findings in their manuscript fully available?

Reviewer #1: Yes

5. Is the manuscript presented in an intelligible fashion and written in standard English?

Reviewer #1: Yes

6. Review Comments to the Author

Reviewer #1: The authors have adequately revised their manuscript, in particular the analysis is more sound and the added explanations and examples provide better intiuition.

7. PLOS authors have the option to publish the peer review history of their article (what does this mean?). If published, this will include your full peer review and any attached files.

Reviewer #1: **Yes: **Kim Fairley

---

## [Editor Report · Acceptance letter]

17 Aug 2021

PONE-D-20-40629R1 

To maximize or randomize?  An experimental study of probability matching in financial decision making 

Dear Dr. Zhang:

I'm pleased to inform you that your manuscript has been deemed suitable for publication in PLOS ONE. Congratulations! Your manuscript is now with our production department. 

Kind regards, 

on behalf of

Dr. Jason Anthony Aimone 

Academic Editor

PLOS ONE